# CLAD: A Contrastive Learning based Method for Multi-Class Anomaly Detection

## Abstract

Anomaly detection is crucial yet challenging in industrial production, especially in multi-class scenarios. Existing high-performance unsupervised methods often suffer from low efficiency and high model complexity. While lightweight discriminator-based detectors have been proposed, they are typically designed for single-class detection and exhibit significant performance degradation when extended to multi-class tasks. To address these limitations, we propose a novel Contrastive Learning-based multi-class Anomaly Detection (CLAD) method. Our approach first encodes multi-class normal images to generate normal samples in the feature space, then synthesizes anomalous samples in this encoded space. We then employ an adapter network to compress the samples and leverage contrastive learning to effectively cluster normal and anomalous samples across multiple classes. Finally, a discriminator network is used for anomaly classification and score prediction. By leveraging anomaly sample generation and a two-stage training process, our framework achieves state-of-the-art performance on the MVTec and VisA datasets under the discriminator-based paradigm. Our key contributions include a novel framework for multi-class anomaly detection, efficient sample generation techniques, and a comprehensive evaluation of model configurations.

## 1 Introduction

Anomaly detection is a critical task in modern industrial production, serving as a key component for ensuring product quality and safety. In practice, detecting anomalies in complex manufacturing processes involves identifying rare, unseen, and often subtle deviations from the expected behavior. Unsupervised anomaly detection methods have gained popularity due to their ability to learn from unlabeled data, making them particularly suitable for real-world applications where obtaining annotated samples is costly and time-consuming. However, the majority of existing approaches are predominantly designed for single-class anomaly detection, which significantly limits their practical applicability in multi-class scenarios where the detection task involves distinguishing between a diverse set of normal and anomalous conditions. Current unsupervised methods can be broadly categorized into two types: reconstruction-based and embedding-based approaches. Reconstruction-based methods, such as UniAD You et al. (2022) and DiAD He et al. (2024), rely on learning to reconstruct normal samples accurately, identifying anomalies based on reconstruction errors. While effective in single-class settings, these methods often struggle with multi-class detection due to their limited generalization capabilities across different normal classes. On the other hand, embedding-based methods like GLASS Chen et al. (2024) and SimpleNet Liu et al. (2023) aim to capture the feature representations of normal samples, yet they are seldom explored in multi-class contexts and often fail to distinguish between complex patterns of normal and anomalous samples.

To bridge this gap, we introduce a novel framework called Contrastive Learning-based Anomaly Detection (CLAD), which leverages contrastive learning to enhance feature embedding and distinguish between normal and anomalous samples across multiple classes. Unlike traditional methods that rely solely on the reconstruction or simplistic feature embedding strategies, CLAD employs a two-stage training process. The first stage utilizes contrastive learning to learn discriminative feature representations by contrasting normal samples against synthesized anomalies, specifically tailored for each class. The second stage refines these learned representations through fine-tuning, effectively adapting the model to multi-class anomaly detection tasks. Our approach addresses two critical challenges in multi-class anomaly detection. First, we tackle the issue of efficiently generating

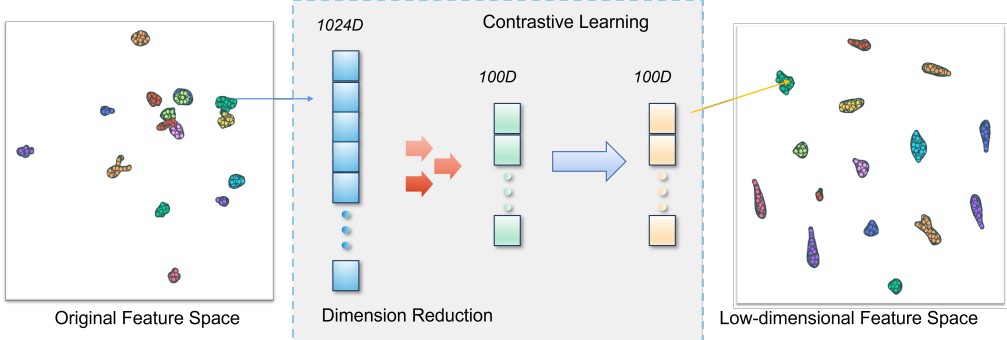

Figure 1: Motiovation. Given the complexity and difficulty of learning multi-class high-dimensional patch feature distributions, which often contain redundant information for anomaly detection, we propose the CLAD method. CLAD focuses on eliminating this redundancy through dimensionality reduction while employing supervised contrastive learning to construct feature distributions with clear inter-class boundaries and compact intra-class structures, making them more suitable for effective anomaly detection.

representative anomaly samples for each class by adapting existing anomaly generation techniques to multi-class settings. This strategy enables the model to learn a robust decision boundary that separates normal and anomalous patterns across different classes. Second, we introduce a supervised contrastive learning strategy that constructs feature distributions based on patch-level class labels, allowing the model to capture fine-grained distinctions between classes.

We conduct extensive experiments on the MVTec Bergmann et al. (2019) and VisA Zou et al. (2022) datasets to validate the effectiveness of CLAD. Our results show that CLAD significantly outperforms state-of-the-art methods in unsupervised multi-class anomaly detection. Key contributions of our work include:

- A novel framework that combines contrastive learning with feature embeddings for robust multi-class anomaly detection.
- An effective anomaly sample generation technique tailored for multi-class scenarios, enhancing the model's ability to learn discriminative features.
- Comprehensive evaluation of model configurations and backbones, providing insights into optimizing anomaly detection performance.

## 2 RELATED WORK

**Multi-class Anomaly Detection.** Current research in multi-class anomaly detection (MUAD) focuses on utilizing diffusion models to generate reference images and applying Vision Transformer (ViT) models for improved performance in multi-class tasks. Specifically, AnoDDPM is one of the earliest models to apply diffusion models for anomaly detection in medical imaging, leveraging their superior image generation and reconstruction capabilities to effectively recover complex features across multiple classes. This allows the model to handle different types of anomalies rather than solely learning features for a single class. Additionally, DiffusionAD enhances anomaly detection through an anomaly synthesis strategy, generating abnormal samples and combining denoising with segmentation networks. However, these methods still face challenges when reconstructing large-scale defects, particularly in handling complex backgrounds or extensive anomalies. DiAD improves this by employing a semantic-level image generation strategy that preserves the semantic information of images, enabling better reconstruction of complex and large anomaly regions, and thereby increasing detection accuracy. This multi-class anomaly detection approach offers new insights into addressing the limitations of existing diffusion models in anomaly localization.

ViT processes the entire image using self-attention mechanisms, capturing long-range dependencies between image patches. This ability allows it to better identify distinguishing features across multiple categories. Traditional anomaly detection methods typically rely on pyramidal structures (multi-resolution) to extract multi-level features, while ViT, with its straightforward architecture, extracts

rich multi-scale features at each layer, making it well-suited for addressing complex anomalies in multi-class scenarios.

For instance, methods like InTra and AnoVit have applied ViT in image reconstruction and anomaly detection, typically capturing image features through ViT encoders to detect anomalies. However, most of these methods utilize standard Transformer architectures and lack in-depth exploration of the specific advantages of Transformers. Recent studies indicate that using a pure ViT architecture for multi-class anomaly detection yields significant performance improvements. For example, DINO pre-trained ViT features perform exceptionally well in multi-class anomaly detection tasks, effectively capturing anomalies of various scales within images, thus reducing information leakage. Furthermore, ViT's hierarchical query decoder allows it to handle both global and local anomalies simultaneously, enhancing robustness and precision in detecting complex anomalies.

Despite their successes, these methods often suffer from complex model designs, resulting in time-consuming inefficiencies. The Runtang Model addresses this issue with a simple contrastive architecture based on convolutional models.

**Efficient Anomaly Detection Methods.** EfficientAD exemplifies the focus on high efficiency rather than single-class detection. This method improves performance through various strategies while maintaining a manageable model complexity.

GLASS integrates Global Anomaly Synthesis (GAS) and Local Anomaly Synthesis (LAS) to synthesize anomalies at both feature and image levels, thereby enhancing detection capabilities across a wider range of anomaly types. GAS uses gradient ascent-guided Gaussian noise for subtle defect detection, while LAS overlays distinct abnormal textures on normal images to manage more pronounced anomalies and increase synthesis diversity.

However, these methods involve complex model designs or optimization strategies. Recent works, such as SimpleNet, present a new direction with a simpler discriminator-based architecture.

SimpleNet combines unsupervised learning and synthetic anomaly generation, using normal samples for training while generating diverse abnormal samples to enhance model performance. This model excels in inference speed and detection accuracy, demonstrating strong adaptability for effective surface defect detection in industrial applications.

**Contrastive Anomaly Detection Methods.** ReConPatch employs a contrastive learning framework to extract patch features from pre-trained models, constructing distinguishable feature representations. By training a linear transformation instead of the entire network, ReConPatch effectively adjusts feature representations, making them more targeted for anomaly detection tasks.

## 3 THE CONTRASTIVE LEARNING BASED METHOD FOR MULTI-CLASS ANOMALY DETECTION

### 3.1 OVERVIEW

The overall architecture of the model is shown in the figure. The model consists of three main components: a feature extractor $E_\Phi$, a dimensional reduction adaptor $A_\phi$, and a discriminator $D_\psi$ with classification capabilities. The training process is divided into three phases: training set preparation, first-stage contrastive learning, and second-stage fine-tuning. Given an AD dataset that contains $N$ classes $\mathbf{C} = \{C_1, C_2, ..., C_N\}$, in one of the class set $X_i = \left\{ (X_{i,normal}^{Train}), (X_{i,normal}^{Test}, X_{i,anomaly}^{Test}) \right\}$. For convenient training, we first use $E_\Phi$ preprocess all images for training into patch features $X_p = E_\Phi(X)$. The images for training contain all classes training set images $X_{i,normal}^{Train}, i \in (1, 2, ..., N)$, and the anomaly images fused by the dtd Cimpoi et al. (2014) dataset images and the foreground zone in all normal images in training datasets. All image class labels are maintained as patch labels, and anomaly patches also have an anomaly class label, the patch labels denoted as $L_p$. The patch features $X_p$ encoded by $A_\phi$ is denoted as $X_A = A_\phi(X_p)$. With both classification and discrimination ability, the patch classifies prediction denoted as $X_C = D_{\phi,c}(X_A)$, the patch anomaly score prediction denoted as $X_S = D_{\phi,s}(X_A)$. We optimize a mixed objective of

$$\mathcal{L} = \mathcal{L}_{contrast}(X_A, L_p) + \mathcal{L}_{var}(X_A, L_p) + \mathcal{L}_c(X_C, L_p) + \mathcal{L}_d(X_S, L_p) \tag{1}$$

Figure 2: The CLAD method framework, consists of three main components: an Encoder $E_\Phi$, a dimensionality reduction Adaptor $E_\Phi$, and a Discriminator $E_\Phi$ with classification capabilities. The Encoder includes a backbone and a patch feature fuser, which is frozen during training. The dataset comprises both anomaly detection datasets and locally anomalous images, as well as globally anomalous features, following the structure of the Glass method. The Adaptor and Discriminator receive patch features for training, where contrastive $\mathcal{L}_{contrast}$ and variance losses $\mathcal{L}_{var}$ are applied to the Adaptor's output, and classification loss $\mathcal{L}_c$ and discriminative $\mathcal{L}_d$ is applied to the Discriminator, which is responsible for the classification and discriminative task.

Where $\mathcal{L}_{contrast}$ is the contrastive loss, $\mathcal{L}_{var}$ is the class distribution variance loss, $\mathcal{L}_c$ is the patch classification loss, $\mathcal{L}_d$ is the patch discriminative loss. In the following, we introduce the details of the model and losses.

## 3.2 Feature extractor and patch feature dataset preparation

Due to the first contrastive learning phase, which can only train on less patch feature batch size, we first preprocess the image dataset to patch feature dataset.

**The patch feature extract process** The feature map for image $x_i \in X_{train}$ extracted by $\Phi$ denoted as $\Phi_{i,j} = \Phi_j(x_i) \in \mathbb{R}^{H_j, W_j, C_j}$. The vector at location $(h, w)$ is represented as $\Phi_{i,j}^{h,w} \in \mathbb{R}^{C_j}$. Similar to PatchCore and Glass, we aggregate the neighborhood features through adaptive average pooling, the locally aware vector $s_{i,j}^{h,w} \in \mathbb{R}^{C_j}$ is obtained from the neighborhood features of $\Phi_{i,j}^{h,w}$ considering a neighborhood size of $p$. The set of vectors $s_{i,j}^{h,w}$ constitutes the feature map $s_{i,j}$. By upsampling to a higher resolution feature map and merging $s_{i,j}$ from different levels, the concatenated feature map $t_i \in \mathbb{R}^{H_m * W_m * C_d}$. The channel size is processed by concat $C = \sum_{j \in J} C_j$ and adopts an adaptive average pooling to destination dimension $C_d$.

**The composition of the patch feature training dataset** In this patch feature training dataset, we not only have the normal images in the training dataset of all classes about the AD dataset but also have the anomaly images synthesis with the DTD Cimpoi et al. (2014) dataset. In this process, we follow the stratege similar GLASS Chen et al. (2024). The synthesized anomaly images fuse from normal images in the training set and DTD textures, with Perlin masks. The Perlin anomaly mask is generated by Perlin noise. With each normal image two Perlin binary masks as $m_1$ and $m_2$, a foreground mask as $m_f$. The final mask is constructed as:

$$m_i = \begin{cases} (m_1 \wedge m_2) \wedge m_f & 0 \le p_m \le \alpha \\ (m_1 \vee m_2) \wedge m_f & \alpha < p_m \le 2\alpha \\ m_1 \wedge m_f & 2\alpha < p_m \le 1 \end{cases} \tag{2}$$

With random number $p_m\ U(0,1)$, $\alpha$ set to $1/3$ in the experiments. The DTD image randomly selected from the DTD dataset will be augmented. The augmentation methods denote as $T = \{T_1, ..., T_K\}$, $K = 9$. In the augmented process, three methods will be chosen to form $T_R \in T$. The augmented texture image is denoted as $x_i'' = T_R(x_i')$. In the fusion process, we adopt a transparency coefficient $\beta \sim N(\mu_m, \rho_m^2)$ to adjust the AD training set image $x_i$ proportion with the synthetic image under the anomaly mask. The local anomaly image $x_{i+}$ is fused as:

$$x_{i+} = x_i \odot \overline{m}_i + (1 - \beta) x_i'' \odot m_i + \beta x_i \odot m_i \tag{3}$$

where $\overline{m}_i$ is derived by inverting the anomaly mask $m_i$. Only the fusion region patch features are extracted and used in the following training process to construct the patch feature training dataset.

### 3.3 THE MODEL FRAMEWORK AND ANOMALY FEATURE SYNTHESIS STRATEGY

In the model framework the feature extractor $E_\Phi$ contracts by backbone and feature fuser, and the dimensional reduction adaptor $A_\phi$ contracts by a three-layer MLP with batch norm between them. The discriminator $D_\psi$ also contrasts by three layers of MLP, the output of discriminator contract by two-part $[X_C, X_S]$, one is the patch classification result in $X_C$, another is the patch anomaly score $X_S$. In the training process, we use the anomaly features extracted from the anomaly images $x_{i+}$ produced by normal images and the dtd textures, and in the training process, we also do another fusion with normal features in the feature-level, which use the same $\beta$ proportion as 3 to fuse the normal features with the anomaly feature extract from $x_{i+}$.

$$X_{pa} = (1 - \beta)x_{dtd} + \beta x_{pn} \tag{4}$$

Except for the local anomaly strategy, we also adopt a global anomaly feature. In this process, we add Gaussian noise on the patch feature $X_p$ extracted from the feature extractor and correct the noise direction to the gradient ascends direction. Same as Glass, in this process we add Gaussian noise on $X_p$ at each dimension with noise $\epsilon \sim N(\mu, \sigma^2)$, denote as $X_{pga} = X_p + \epsilon$. For effective training, we will correct the noise direction to the gradient ascent direction, as:

$$\overline{X}_{pga} = X_{pga} + \eta \frac{\nabla L_{gas}(X_{pga})}{||L_{gas}(X_{pga})||} \tag{5}$$

$$L_{gas} = \sum f_{BCE}(X_{pga}, 1) \tag{6}$$

To project $X_{pga}$ onto the set $N_p = X_{pga}|r_1 < ||X_{pga} - X_p||_2 < r_2$, the gradient ascent distance is $\overline{\epsilon} = \overline{X}_pga - X_p$, the truncated distance $\hat{\epsilon}$ is given by:

$$\hat{\epsilon} = \frac{\alpha}{||\overline{\epsilon}||}\overline{\epsilon}, \; where \; \alpha = \begin{cases} r_1 & ||\overline{\epsilon}|| < r_1 \\ r_2 & ||\overline{\epsilon}|| > r_2 \\ ||\overline{\epsilon}|| & otherwise \end{cases} \tag{7}$$

Finally, the global anomaly feature $X_{pga} = X_p + \hat{\epsilon}$.

### 3.4 CONTRASTIVE LEARNING METHOD AND TRAINING OBJECTIVES

In this method, we hope to eliminate the noise in the feature, with no help in anomaly detection, through the decline of the feature dimension. As the multi-class feature distribution is complicated, we also hope to separate the feature distributions into different classes. So we introduce the supervised contrastive learning aim to separate different classes distribution in the hidden space encoded by the dimensional reduction adaptor $A_\phi$. With the contrastive learning target $X_A$, the batch size is denoted as $B$, For each pair of samples $i$ and $j$, where $i, j \in \{1, 2, \ldots, B\}$, the Euclidean distance between their means is computed, resulting in a $B * B$ distance matrix:

$$D_{\text{mean}}(i, j) = ||\mu_i - \mu_j||_2, \quad i, j \in \{1, 2, \ldots, B\} \tag{8}$$

Based on the labels, we calculate a $B \times B$ matrix, which indicates whether the samples in the pair have the same label:

$$label\_equal(i, j) = \begin{cases} 1, & \text{if label}_i = \text{label}_j \\ 0, & \text{if label}_i \neq \text{label}_j \end{cases}, \quad i, j \in \{1, 2, \ldots, B\} \tag{9}$$

$$label\_not\_equal(i, j) = 1 - \text{label\_equal}(i, j) \tag{10}$$

For positive sample pairs (those with the same label), we compute the Euclidean distance between them and use the normal sample indicator $is\_normal$ and the label equality indicator $label\_equal$ to weigh the distances. The positive sample pair loss is defined as the distance between the samples multiplied by these indicators:

$$\mathcal{L}_{pos}(i, j) = D_{mean}(i, j) \cdot label\_equal(i, j), \quad i, j \in \{1, 2, \ldots, B\} \tag{11}$$

For negative sample pairs (those with different labels), we compute the Euclidean distance and apply a margin $M$ (distance threshold) to ensure that smaller distances between negative pairs receive a higher penalty. The negative sample pair loss is calculated as the difference between the margin and the pairwise distance, multiplied by the indicators for label inequality.

$$\mathcal{L}_{neg}(i,j) = \max(0, M - D_{mean}(i,j)) \cdot label\_not\_equal(i,j), \quad i,j \in \{1,2,\ldots,B\} \tag{12}$$

To emphasize more difficult sample pairs, we compute weights for the positive and negative sample pairs using an exponential function. The weight formulas $w_pos$ and $w_neg$ depend on the loss values and are adjusted by the hyperparameters $\alpha$ and $\gamma$:

$$w_{pos}(i,j) = \alpha \cdot (1 - \exp(-\mathcal{L}_{pos}(i,j)))^{\gamma}, \quad i,j \in \{1,2,\ldots,B\} \tag{13}$$

$$w_{neg}(i,j) = \alpha \cdot (1 - \exp(-\mathcal{L}_{neg}(i,j)))^{\gamma}, \quad i,j \in \{1,2,\ldots,B\} \tag{14}$$

The calculated weights $w_pos$ and $w_neg$ are applied to the positive and negative sample pair losses, resulting in the weighted positive and negative sample pair losses:

$$\mathcal{L}_{weighted\_pos}(i,j) = w_{pos}(i,j) \cdot \mathcal{L}_{pos}(i,j), \quad i,j \in \{1,2,\ldots,B\} \tag{15}$$

$$\mathcal{L}_{weighted\_neg}(i,j) = w_{neg}(i,j) \cdot \mathcal{L}_{neg}(i,j), \quad i,j \in \{1,2,\ldots,B\} \tag{16}$$

Finally, the weighted losses for all sample pairs in the batch are summed and averaged by dividing by $B^2$, producing the average loss, which serves as the:

$$\mathcal{L}_{contrast} = \frac{1}{B^2} \sum_{i=1}^{B} \sum_{j=1}^{B} (\mathcal{L}_{\text{weighted\_pos}}(i,j) + \mathcal{L}_{\text{weighted\_neg}}(i,j)) \tag{17}$$

In the function $\mathcal{L}_{var}$, a variance regularization term is introduced to designed to reduce intra-class variance in the learned feature space. Let $\mu_i$ represent the mean vector of class $i$, indicating the feature center of the class. The variance of the sample features for each class $i$ can be expressed as:

$$\sigma_i^2 = \frac{1}{N_i} \sum_{j=1}^{N_i} \left( z_j^{(i)} - \mu_i \right)^2 \tag{18}$$

where $N_i$ is the number of samples in class $i$, and $z_j^{(i)}$ denotes the feature of the $j$-th sample belonging to class $i$. The overall variance regularization loss is the average of variances across all classes:

$$L_{\text{variance}} = \frac{1}{C} \sum_{i=1}^{C} \sigma_i^2 \tag{19}$$

where $C$ is the number of classes. This regularization encourages the feature representations of samples within the same class to be more compact, thereby improving intra-class consistency.

The classification loss function $\mathcal{L}_c(X_C, L_p)$ utilizes the Cross-Entropy Loss, which is commonly used in classification tasks. After $X_A$ passes through $D_\psi$, the logits for classification will obtained. The cross-entropy loss is then computed between the predicted logits and the true labels, with $C$ as the number of classes, making it suitable for standard classification problems where we need to distinguish between normal and anomalous classes.

$$L_{CE} = - \sum_{i=1}^{B} \sum_{c=1}^{C} L_{P_{i,c}} \log(X_{C_{i,c}}) \tag{20}$$

The Hinge loss function is designed specifically for anomaly detection using a hinge loss approach. In this case, anomalous samples (labeled as 15) and normal samples are treated separately:

For the anomaly detection case, the hinge loss is computed separately for anomalous and normal samples. For anomalous samples $z_i$, we want the anomaly score $s_i$ to stay below a threshold $\delta$, while for normal samples, the score should be higher than $-\delta$. The hinge loss for anomalous samples is:

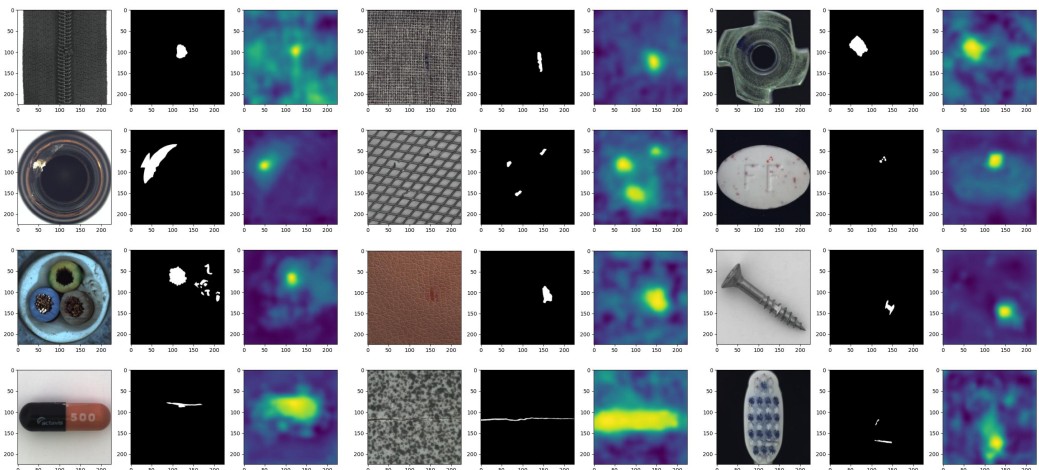

Figure 3: The effect picture. Because of intricate multi-class distribution in high dimensions, is hard to learn, and redundant information is unnecessary for anomaly detection. We propose the CLAD method, which decreases the patch feature dimension and uses surprised contrastive learning to formulate a distinct separate and compact distribution
.

$$L_{anomalous} = \max(0, \delta - s_i) \tag{21}$$

For normal samples, the hinge loss is:

$$L_{normal} = \max(0, s_i + \delta) \tag{22}$$

The total hinge loss is the mean of both losses:

$$L_{Hinge} = \frac{1}{N} \left( \sum_{i \in anomalous} L_{anomalous} + \sum_{i \in normal} L_{normal} \right) \tag{23}$$

### 3.5 TWO-STAGE TRAINING PROCESS

The introduction of contrastive learning has led to a significant decrease in the number of features that can be trained simultaneously. Therefore, after the contrastive learning phase, we believe that the feature distribution in the space outputted by the Adaptor is already relatively reasonable, and the presence of the contrastive learning loss would reduce training efficiency. Thus, we choose to disable the contrastive learning loss in the second stage while retaining the other losses:

$$\mathcal{L} = \mathcal{L}_c(X_C, L_p) + \mathcal{L}_d(X_S, L_p) \tag{24}$$

Experimental results indicate that through this approach, our algorithm achieves current state-of-the-art performance. Further ablation experiments confirm the effectiveness of each component in the process.

## 4 EXPERIMENT

### 4.1 SETUPS FOR MULTI-CLASS UNSUPERVISED AD

**Task Setting.** This work focuses on training all classes in the AD dataset, with none of the truth anomaly images reachable. In the inference process, we use only one model to detect all classes of normal or anomalous images. Both anomaly detection and localization are required.

**MVTec-AD dataset.** The MVTec-AD dataset Bergmann et al. (2019) is designed for unsupervised anomaly detection in industrial scenarios. It contains 5,354 high-resolution images across 15 categories (5 textures, 10 objects) from various industrial domains. The data set is divided into a training set of 3,629 anomaly-free images and a test set of 1,725 images with normal and abnormal samples. In the test set, pixel-level annotations for anomalies are provided, allowing the evaluation of both detection and localization tasks. This comprehensive dataset fills a critical gap in industrial anomaly detection research, offering a standardized benchmark for algorithm development and assessment in realistic production environments.

**VisA dataset.** The VisA (Visual Anomaly) dataset, introduced by Zou et al. (2022), is a comprehensive resource for visual anomaly detection research. It comprises 10,821 high-resolution images, including 9,621 normal and 1,200 anomalous samples featuring 78 types of anomalies. The dataset is structured into 12 subsets, each representing a distinct object. These objects are categorized into complex structures, multiple instances, and single instances. This diverse composition allows for a thorough evaluation of anomaly detection algorithms in various complexities and scenarios of objects. VisA is valuable for developing and testing robust visual inspection methods in industrial and research applications.

**Evaluation Metrics for AD.** Similar to (Deng & Li, 2022; Zavrtanik et al., 2021; Bergmann et al., 2020), we use threshold-independent measures, including mean Area under the Receiver Operating Curve (mAU-ROC) to evaluate binary classification ability and mean Area Under the Per-Region-Overlap Bergmann et al. (2020) (mAU-PRO) to weigh regions of different sizes equally. Note that mAU-ROC is used in image-level (anomaly classification) and pixel-level (anomaly segmentation) evaluations. The maximum pixel-level value is regarded as the image-level anomaly score Deng & Li (2022); You et al. (2022). The models are evaluated ten times evenly for all methods, and the result corresponding to the maximum pixel-level mAU-ROC value is taken as the final result. We demonstrate and emphasize using all metrics for evaluation.

**Comparision Methods.** As MUAD is a relatively new task, we mainly evaluate the published UniAD You et al. (2022) methods. We also compare with the latest augmentation-based DRAEM Zavrtanik et al. (2021), reconstruction-based RD Deng & Li (2022), and Embedding-based SimpleNet Liu et al. (2023). Since the above methods only report results under the SUAD setting, we retrain them to obtain MUAD results by official codes.

**Training.** In this study, we employ WideResNet50 as the backbone network for CLAD. The input images are resized to 256x256, followed by center-cropping to 224x224, without applying any data augmentation. We use the AdamW optimizer with a learning rate set to 0.0002. The training is divided into two stages: In the first stage, a batch size of 8000 patch features is used, and the model is trained for 200 epochs. The best-performing model from this stage is then fine-tuned in the second stage, with a batch size of 100,000 patch features for 2000 epochs. All images are trained together, using their labels during training but not during testing. The backbone remains frozen throughout the training process. The entire experiment is conducted on dual 3090 GPUs. $\beta$ is set to 0.5, $\gamma$ is set to -0.8.

## 4.2 COMPARATIVE EXPERIMENTS ON DIFFERENT DATASETS

We evaluate the CLAD method with state-of-the-art approaches using both image-level and pixel-level metrics (see Table 1) on the MVTec AD dataset. The proposed CLAD method performs favorably against all the evaluated schemes. CLAD achieves better image-level results than DiAD with SoTA results on mAU-ROCsp/mAU-ROCpx/mAUPROpx 97.5/97.0/96.0. In addition, CLAD achieves a performance gain of +0.3 ↑/+0.2 ↑/+5.3 ↑ using the mean metric.

We have a few findings from these empirical results. First, classifying information can help model learning and improve performance. Second, an appropriate compact class distribution and a clear distinction between different class distributions can benifit the performance. Third, diverse anomaly and tiny can also help to have better performance.

The VisA dataset contains more complex structures, multiple and large variations of objects, and more images. The quantitative results in (see Table 2) show that CLAD consistently performs well against state-of-the-art schemes. CLAD surpasses UniAD by mAU-ROCsp/mAU-ROCpx/mAUPROpx of +2.2 ↑/-2.2 ↑/+9.4 ↑, and show the potential of CLAD.

| Category | CFLOW-AD | SimpleNet | RD | UniAD | DiAD | CLAD |
|---|---|---|---|---|---|---|
| bottle | 99.9/97.3/92.2 | **100.0**/97.6/90.1 | 99.7/97.8/94.8 | 99.8/**98.1**/95.3 | 99.7/**98.4**/- | **100.0**/97.0/**96.0** |
| cable | 90.8/89.9/79.4 | 99.0/96.7/87.3 | 88.2/84.9/78.9 | 96.6/**97.0**/86.6 | 94.8/96.8/- | **98.2**/95.4/**93.8** |
| capsule | 87.8/98.5/93.4 | 98.3/98.4/96.0 | **98.3/98.8**/96.0 | 87.5/98.7/92.5 | 89.0/97.1/- | 91.3/98.2/**97.9** |
| carpet | 99.4/98.9/94.7 | 97.0/98.9/91.3 | 99.0/**99.0**/95.9 | **99.9**/98.6/95.2 | 99.4/98.6/- | 98.9/98.2/**97.6** |
| grid | 89.4/93.6/82.5 | 96.4/96.1/87.1 | **99.2/99.3**/97.6 | **99.2**/97.0/92.1 | 98.5/96.6/- | 97.5/96.8/95.7 |
| hazelnut | **100.0**/98.6/95.8 | **100.0**/98.3/93.3 | **100.0/98.7/96.5** | 99.9/98.3/94.6 | 99.5/98.3/- | **100.0**/97.9/96.7 |
| leather | **100.0**/99.3/98.3 | **100.0**/99.2/95.3 | **100.0/99.4/98.1** | **100.0**/99.1/97.6 | 99.8/98.8/- | **100.0**/98.5/98.0 |
| metal nut | 98.0/96.0/88.8 | 98.7/97.9/92.1 | **99.8**/94.4/92.4 | 98.5/93.4/80.9 | 99.1/97.3/- | 99.4/**97.4/96.7** |
| pill | 85.1/96.5/90.9 | 91.5/96.4/85.3 | **98.6/97.5**/96.1 | 94.2/95.1/94.7 | 95.7/95.7/- | 92.2/97.8/**97.5** |
| screw | 71.6/97.0/89.3 | 81.8/96.3/86.8 | **98.3/99.4/97.2** | 92.4/98.9/94.2 | 90.7/97.9/- | 91.4/97.5/97.1 |
| tile | 99.8/96.0/86.8 | **99.9**/96.6/83.1 | 99.4/95.3/86.2 | **100.0**/92.6/81.6 | 96.8/92.4/- | 99.6/95.1/**93.0** |
| toothbrush | 83.9/98.2/85.8 | 91.7/98.2/81.2 | 99.2/**99.0**/93.0 | 90.3/98.6/87.9 | **99.7/99.0**/- | 96.9/97.8/**97.1** |
| transistor | 92.5/84.6/74.1 | 98.2/94.5/82.6 | 94.8/85.6/74.8 | **100.0/97.7/94.4** | 99.8/95.1/- | 98.6/96.5/92.5 |
| wood | 98.9/94.4/91.0 | **99.9**/95.6/80.2 | 99.6/**95.6/92.0** | 98.8/93.7/89.6 | **99.7**/93.3/- | 99.5/92.8/90.5 |
| zipper | 96.2/98.1/93.1 | 99.8/**99.7**/95.6 | 99.8/98.5/95.6 | 95.3/97.0/91.4 | 95.1/96.2/- | **99.6**/98.2/**97.7** |
| Avg | 92.7/95.8/89.0 | 95.4/96.7/87.6 | 96.9/95.9/92.0 | 96.8/96.8/91.0 | 97.2/96.8/90.7 | **97.5/97.0/96.0** |

Table 1: Comparison with SOTA methods on MVTec-AD dataset for multi-class anomaly detection with $mAUROCspmax(Max)/mAUROCpx(Max)/mAUPROpx(Max)$ metrics.

| Category | CFLOW-AD | SimpleNet | UniAD | DiAD | CLAD |
|---|---|---|---|---|---|
| candle | 93.7/99.2/**96.7** | 95.9/**97.7**/92.5 | 97.7/**99.3**/94.8 | 92.8/97.3/89.4 | 97.5/97.39/96 |
| capsules | 57.8/94.6/81.4 | **77.0**/95.9/70.0 | 73.8/**98.4**/81.2 | 58.2/97.3/77.9 | 72/95.3/**95** |
| cashew | **96.3/99.1**/94.8 | 93.4/98.7/85.0 | 93.4/99.0/91.4 | 91.5/90.9/61.8 | 93.6/98.4/**97.8** |
| chewinggum | 97.5/99.2/94.9 | 98.1/98.3/83.1 | **99.2/99.3**/88.4 | 99.1/94.7/59.5 | 97.3/97.2/**96.2** |
| fryum | 92.5/**97.6/94.9** | 87.3/96.5/83.3 | 91.1/97.3/85.5 | 89.8/97.6/81.3 | **95.6**/90.9/88.8 |
| macaroni1 | 82.0/97.7/95.2 | 79.9/97.7/90.9 | 88.1/**99.4**/95.9 | 85.7/94.1/68.5 | **91.6**/97.6/**96.7** |
| macaroni2 | 67.1/97.5/**95.0** | 67.8/92.9/84.7 | **81.5/98.3**/92.9 | 62.5/93.6/73.1 | 73.5/92.3/91.7 |
| pcb1 | 94.9/99.2/**96.9** | 92.5/98.8/81.1 | **96.3/99.4**/90.5 | 88.1/98.7/80.2 | 91.8/97.6/96.7 |
| pcb2 | 92.8/96.7/88.1 | **94.1**/97.5/84.2 | 93.7/**98.4**/86.3 | 91.4/95.2/67.0 | 92.9/95.2/**93.2** |
| pcb3 | 81.5/96.4/91.1 | 89.4/97.8/83.7 | 90.0/**98.5**/86.2 | 86.2/96.7/68.9 | **91.4**/97/**95.6** |
| pcb4 | 98.9/96.8/85.6 | 98.6/96.6/82.8 | 99.4/**97.6**/85.3 | **99.6**/97.0/85.0 | 98.7/95.6/**92.9** |
| pipefryum | **97.8**/99.2/97.0 | 87.6/99.1/83.8 | 97.0/99.0/94.0 | 96.2/**99.4**/89.9 | 96/98.9/**98.4** |
| Avg | 87.2/97.8/94.8 | 87.7/96.9/82.4 | 88.8/**98.3**/85.5 | 86.8/96.0/75.2 | **91**/96.1/**94.9** |

Table 2: Comparison with SOTA methods on VISA dataset for multi-class anomaly detection with $mAUROCspmax(Max)/mAUROCpx(Max)/mAUPROpx(Max)$ metrics.

## 5 CONCLUSION AND LIMITATION

In this work, we are the first to introduce a feature embedding-based discriminative approach into multi-class anomaly detection. By leveraging dimensionality reduction and contrastive learning, we propose the CLAD method, which achieves state-of-the-art performance using only an MLP. Our results demonstrate the effectiveness of feature embedding-based approaches, offering a novel perspective distinct from reconstruction-based multi-class anomaly detection methods. This provides a new pathway for advancing the development of multi-class anomaly detection algorithms.

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
