# OpenReview forum: "CLAD: A Contrastive Learning based Method for Multi-Class Anomaly Detection"
_ICLR.cc/2025/Conference — ICLR 2025 Conference Withdrawn Submission_

### Official Review · Reviewer_evHN · 2024-10-24

**Soundness:** 2
**Presentation:** 2
**Contribution:** 2
**Rating:** 1
**Confidence:** 5

**Summary:**

This paper proposes to enhance existing embedding-based anomaly detection methods to multi-class anomaly detection through contrastive learning. There exists several critical problems thus I vote to reject this paper.

**Strengths:**

The proposed method achieves better anomaly detection performance than the selected alternatives.

**Weaknesses:**

Related works are not comprehensive enough, which makes this manuscript less convincing. The authors categorize anomaly detection methods into reconstruction-based and embedding-based methods, ignoring lots of related works like memory-bank-based, knowledge-distillation-based, etc.

Also, lots of works lack proper references, like the author mentions DINO without refer to the corresponding paper.

This paper also lacks innovations. The authors utilize several existing techniques like gradient ascending from GLASS to construct anomalies in the feature space.

GLASS is quite an important baseline for this paper, but the authors didn’t compare the proposed method to GLASS.

There are even no ablation studies.

**Questions:**

NA

---

### Official Review · Reviewer_gFUN · 2024-11-01

**Soundness:** 1
**Presentation:** 1
**Contribution:** 1
**Rating:** 3
**Confidence:** 5

**Summary:**

The authors propose a multi-class anomaly detection method based on contrastive learning. They introduce a two-stage training framework to train the Adapter and Discriminator separately. The effectiveness of the method is validated on the anomaly detection metrics of the MVTec-AD and VisA datasets.

**Strengths:**

The authors propose a multi-class anomaly detection method based on contrastive learning. They introduce a two-stage training framework to train the Adapter and Discriminator separately.

**Weaknesses:**

See Questions.

**Questions:**

1. The paper lacks ablation studies, making the overall structure incomplete.
2. In lines 012-013 of the abstract, the authors claim to address the high computational complexity of existing models with their proposed framework. However, there are no subsequent experiments to support or validate this claim.
3. The experimental results do not show a significant advantage over existing multi-class anomaly detection methods. It is recommended to validate the method's effectiveness on more datasets.
4. Since there is already a contrastive learning-based anomaly detection method, ReConPatch, the authors should compare their method with ReConPatch in the experiments.

---

### Official Review · Reviewer_7Hic · 2024-11-02

**Soundness:** 1
**Presentation:** 1
**Contribution:** 2
**Rating:** 3
**Confidence:** 5

**Summary:**

The proposed method utilizes the anomalies generated following the GLAS method and incorporating additional losses and training techniques to improve the results. For training, the contrastive losses, hard negative mining, and a variance regularization term are used. The method is evaluated on the MVTec AD and the VisA datasets. The proposed method achieves excellent results on both datasets.

**Strengths:**

- Good performance on the multi-class setting.
- The proposed method is generally well described.

**Weaknesses:**

- Lacking related work (only 10 papers are cited from the very active field of anomaly detection)
- Severly lacking evaluation section. No ablation study. One of the contributions listed in the introduction is a thorough evaluation of model backbones and configurations which is not fulfilled in the rest of the paper.

While the proposed method achieves solid performance on the multi-class setting, the evaluation section is  severely lacking. The design choices, such as the additional loss terms, are not properly evaluated in an ablation study which makes it impossible for the reader to extract important insights and understand the contribution of individual components. This can not be overseen and would require a rewrite of a significant part of the paper to include so I believe this alone is grounds for rejection.

**Questions:**

Related work is lacking. In total 10 papers are cited but the anomaly detection field is very broad. Even the competing methods that the proposed method is compared with in the evaluation are not cited.

At times poorly written - Section 3.3 is an example. The first few sentences are very difficult to understand. Also lines 231 to 247 could be rewritten to improve clarity since it is filled with typos and unclear notation.

L323 – The anomalous examples should be higher than sigma, the normal samples should be lower than minus sigma according to equations 21 and 21. Incorrectly stated in the text if I am not mistaken, although this is a minor detail.

The evaluation section is severely lacking. An ablation study evaluating design choices such as the use of the GLAS-based anomaly generation method, weighted loss, use of the contrastive objective, use of L_var etc.

---

### Official Review · Reviewer_jM1K · 2024-11-03

**Soundness:** 1
**Presentation:** 1
**Contribution:** 1
**Rating:** 1
**Confidence:** 4

**Summary:**

The paper introduces a framework for robust multi-class anomaly detection. It first generates representative anomaly samples for each class by adapting existing anomaly generation techniques. A supervised contrastive learning strategy is then introduced to construct feature distributions based on patch-level class labels, allowing the model to capture fine-grained distinctions between classes. The method achieves state-of-the-art results on various benchmarks, demonstrating both its effectiveness and efficiency.

**Strengths:**

It proposed a framework for robust multi-class anomaly detection.

**Weaknesses:**

1.The paper's writing quality is below standard, with excessive notation, numerous typos, and poor organization, all of which significantly hinder readability.

2.The paper presents only the main results and lacks additional experiments, such as ablation studies, sensitivity analyses, etc., which are essential for thoroughly evaluating the robustness and effectiveness of the proposed approach.

3.Overall, I believe this is an incomplete work that requires further development and additional experiments to fully validate the findings. It should be withdrawn in its current version.

**Questions:**

1.What does $p_m$ U (0, 1) mean in Line 209? What is $x_i′$ in Line 212?

2.What is $\mu$ in Eq.8? Where is the equation for calculating anomaly score $s_i$ in Line 323? You did not mention it in the methodology.

3.Is the batch size B in contrastive learning equal to the number of multi-class categories? How do you balance the computational cost and training efficiency when dealing with a large number of categories?

4.See weakness above.

---

### Official Review · Reviewer_2qUA · 2024-11-03

**Soundness:** 2
**Presentation:** 2
**Contribution:** 2
**Rating:** 3
**Confidence:** 4

**Summary:**

The paper introduces a  Contrastive Learning-based multi-class Anomaly Detection (CLAD) method designed to enhance multi-class anomaly detection in industrial contexts. The proposed method combines dimensionality reduction and supervised contrastive learning to generate and differentiate between normal and anomalous samples across various classes. It uses a two-stage training process involving initial discriminative feature learning and subsequent fine-tuning, significantly improving detection accuracy on benchmark datasets like MVTec and VisA.

**Strengths:**

+  The method effectively leverages supervised contrastive learning to enhance feature separability across multiple classes.
+ The use of a dimensionality reduction adaptor helps reduce feature complexity while preserving relevant information for anomaly detection, making the model more efficient and focused on essential patterns.

**Weaknesses:**

+ The author has made an error in the plotting of Figure 2, as the elements within the figure are not aligned properly.
+ The paper lacks ablation studies and could be expanded to include a deeper analysis of how each training loss term contributes to the final model performance.
+ The layout of tables and text in the experimental section of the article is not rigorous, with excessive spacing.
+ The model structure in the article is simple and lacks innovation.
+ The paper has a limited number of citations and lacks a thorough comparison with relevant existing work.

**Questions:**

The writing quality of the article is subpar, and the paper appears to lack certain crucial elements: there are inaccuracies in the referencing of figures, particularly early in the overview section; there is a notable absence of analysis pertaining to experimental and visualization outcomes; the paper does not include any ablation studies; and there is a scarcity of citations and comparative work. Does this suggest that the work is yet to be considered complete?

---

### Note · Authors · 2024-11-14

I have read and agree with the venue's withdrawal policy on behalf of myself and my co-authors.